October 4, 2016

# Retraction Notice

Retraction: Polwiang S. (2015) The seasonal reproduction number of dengue fever: impacts of climate on transmission. PeerJ 3:e1069 https://doi.org/10.7717/peerj.1069

The author wishes to Retract this article. Some of the original data were supplied by the Thailand Bureau of Vector-Borne Disease, however at that time an incorrect assumption was made by the author when converting weekly data into monthly data. As a result, the number of 226 cases per 100,000 which is used in the model is incorrect. In addition, the figure of 0.5 used for the "infection rate in mosquito's egg" ($\gamma$ in equations 4a and 4b) is also incorrect (the author believes the figure should have been 0.15). These errors mean that many of the specific numerical results in the paper are in error, although the author believes that the overall conclusions are still correct and that the general shape of the graphs remains unchanged (albeit with different values). Because of these errors and the fact that the correct values for the input data are uncertain, the article is Retracted. Interested readers should note that the author has used the current best estimates for the correct data to update a preprint of this paper at https://peerj.com/preprints/756v5/.

Polwiang S. 2016. Retraction: The seasonal reproduction number of dengue fever: impacts of climate on transmission. PeerJ 3:e1069/retraction https://doi.org/10.7717/peerj.1069/retraction



# The seasonal reproduction number of dengue fever: impacts of climate on transmission

Sittisede Polwiang

Department of Mathematics, Faculty of Science, Silpakorn University, Thailand

## ABSTRACT

**Background.** Dengue fever is a mosquito-borne viral disease and a regular epidemic in Thailand. The peak of the dengue epidemic period is around June to August during the rainy season. It is believed that climate is an important factor for dengue transmission.

**Method.** A mathematical model for vector–host infectious disease was used to calculate the impacts of climate to the transmission of dengue virus. In this study, the data of climate and dengue fever cases were derived from Chiang Mai during 2004–2014, Thailand. The value of seasonal reproduction number was calculated to evaluate the potential, severity and persistence of dengue infection.

**Results.** The mosquito population was increasing exponentially from the start of the rainy season in early May and reached its the peak in late June. The simulations suggest that the greatest potential for the dengue transmission occurs when the temperature is 28.9 °C. The seasonal reproduction numbers were larger than one from late March to end of August and reaching the peak in June. The highest incidences occurred in August due to the delay of transmission humans-mosquito-humans. Increasing mean temperature by 1 °C, the number of incidences increases 28.1%. However, a very high or very low temperature reduces the number of infection.

**Discussion and Conclusion.** The results show that the dengue infection depends on the seasonal variation of the climate. The rainfall provides places for the mosquitoes to lay eggs and develop to the adult stage. The temperature plays an important role in the life cycle and behavior of the mosquitoes. A very high or very low temperature reduces the risk of the dengue infection.

## INTRODUCTION

Dengue fever is the most frequent mosquito-borne viral disease in the humans and has become a major international public health concern in recent decades. Over 50 millions people living in tropical and subtropical urban areas from Latin America to South East Asia are infected with dengue virus annually. Infected individuals may be asymptomatic or have high fevers, headache, muscle and joint pain, and a characteristic skin rash similar to measles. These symptoms can develop into the life-threatening dengue hemorrhagic fever and into the dengue shock syndrome. The dengue fever is caused by one of the four

Corresponding author
Sittisede Polwiang,
polwiang_s@su.ac.th

distinct serotypes of dengue virus (DENV), DENV1-4 (*World Health Organization, 2014*). A recovery from the infection by one serotype provides lifelong immunity against that particular serotype but confers only temporary immunity against the other serotypes for approximately 2 years (*Montoya, Gresh & Mercado, 2013*). For dengue virus, the infection is transmitted through an intermediate vector: infected mosquitoes. The primary vector of DENV is *Aedes aegypti* and the secondary vector is *Aedes albopictus*. Aedes mosquitoes are found throughout the tropical and subtropical areas, and they have adapted to cohabiting with humans in both the urban and the rural environment. *Aedes aegypti* bite primarily during the day and are most active for approximately two hours after sunrise and several hours before sunset, but can bite at night in well-lit areas. Only females bite to obtain blood in order to gain nutrients for eggs laying (*Centers for Disease Control and Prevention, 2014*).

Thailand is a tropical country, with a relatively high temperature and humidity all year-round. These conditions are ideal for Aedes mosquitoes to establish themselves. Dengue fever is a local epidemic in Chiang Mai, Thailand, throughout the year, with the endemic period from June to August (*Campbell et al., 2013*). The counter dengue programs provided by public health services consist of educating people how to remove the breeding sites of mosquitoes inside and outside residential areas, preventing mosquitoes from biting, and controlling the mosquito population during peak dengue season. Statistically data suggested that these programs are still unable to stop the disease. The number of dengue fever incidences is increasing. All four dengue serotypes have been detected (*Anantapreecha et al., 2005*). Major dengue outbreaks have occurred irregularly every 3–4 years.

The environment, climate variables such as temperature, humidity, season and rainfall significantly influence the mosquito development. Several studies suggest that entomological parameters are temperature sensitive as the dengue fever normally occurs in tropical regions (*Liu-Helmersson et al., 2014*). The high temperature increases the lifespan of mosquitoes and shortens the extrinsic incubation period of the dengue virus, thereby increasing the number of infected mosquitoes (*Wu, Lay & Guo, 2009*). The rainfall provides places for eggs and for larva development (*Chompoosri et al., 2012*). The climate change will certainly affect the abundance and distribution of dengue vectors (*Khasnis & Nettleman, 2005*). Exploring the relationships between the climate and the dengue transmission is an important task.

In recent decades, mathematical models were developed to investigate the infectious epidemiology. Most of the models incorporate several factors of the disease to predict the possible magnitude of the outbreaks. The basic reproduction number, $R_0$, is defined as the number of infected people generated by a single infectious person in an entirely susceptible population during infectious period. Typically, if $R_0 > 1$ an epidemic occurs while $R_0 < 1$, indicates no outbreak. The larger value of $R_0$ means the harder to control the epidemic (*Heffernan, Smith & L M, 2005*). Estimations of the reproduction number of dengue fever have varied widely. This suggests highly heterogeneous levels of population immunity, as well as vector density coupled with weather conditions.

The objectives of this study were (i) to improve knowledge of the relationships between the climate sensitive variables and the dengue transmission dynamics, (ii) to identify

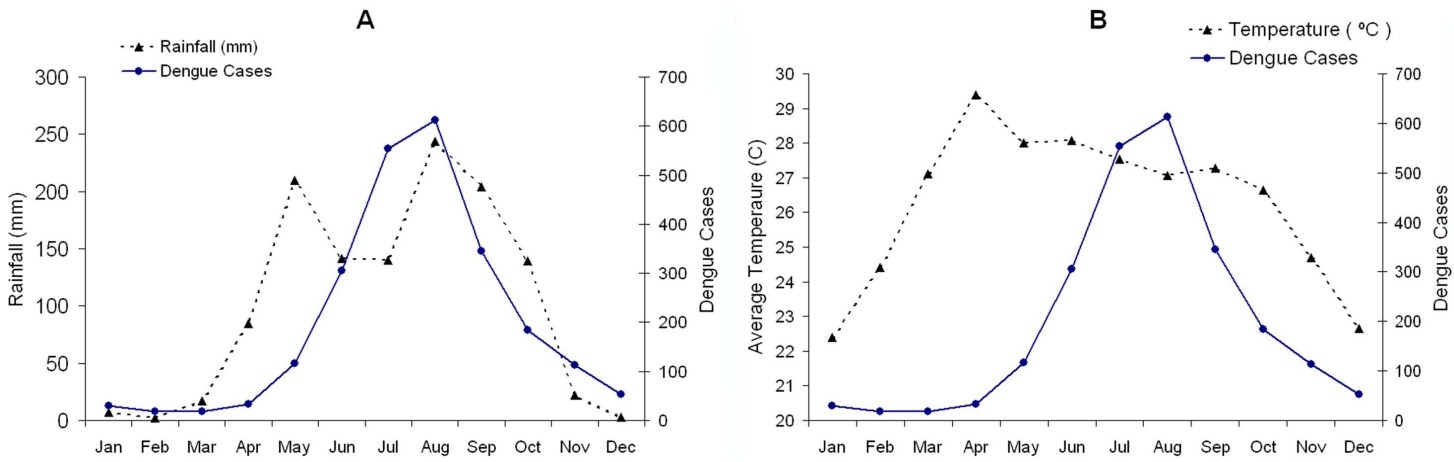

**Figure 1** **The average monthly incidence rate of dengue during 2004–2014 in Chiang Mai, Thailand is indicated by the complete line.** The broken lines are (A) the average monthly rainfall and (B) the mean temperature during the same period.

optimal conditions for a dengue epidemic potential, and (iii) to develop a model for dengue transmission.

## THEORY AND METHODS

### The dengue situation in Chiang Mai, Thailand

This study was conducted using a mathematical model based on the demographic information of Chiang Mai province, Thailand. Chiang Mai is the largest province in the northern Thailand, with a population of approximately 1.6 millions. The mean daily temperature is 22–29 °C, and the rainy season starts in May and last until September. Dengue fever is one of the major public health concern in Chiang Mai. According to the Thailand Bureau of Vector-Borne Disease (*2014*), the largest outbreak in the last decade was reported in 2013 with 11,432 cases, compared to 664 in 2006 and 2,733 on average, which is 226 cases per 100,000 people per year during 2004–2014. Figure 1 shows the average monthly dengue incidences in Chiang Mai from 2004 to 2014 with (A) the average monthly rainfall during the same period and (B) the average monthly temperature (*Meteorological Department of Thailand, 2014*). The average dengue incidences range from 18 in February to 612 in August (*Bureau of Vector-Borne Disease, Department of Disease Control, 2014*). It is evident in the figure that the dengue incidences and the amount of rainfall are well correlated. The correlation between the dengue fever and the rainfall could be explained by increases in adult survival, breeding sites for eggs and feeding activity of the mosquitoes. Humidity increases the oviposition rate and extends the life span of adult mosquito (*Canyon, Hii & Müller, 1999*). However, the available data relationship between the humidity and the dengue transmission parameters is still insufficient to allow for a precise mathematical description.

### Temperature dependent parameters

There are several parameters of the dengue transmission and mosquito life cycle that are temperature sensitive ($T$). Our approach is based on scientific literature on the dengue

transmission with climate sensitive and vector parameters. Several works have shown that such a relationship exist and that it can be described by means of mathematical equations (*Liu-Helmersson et al., 2014*).

### *Vector–host transmission parameters*

Transmission processes involve the infected mosquitoes biting humans after the virus incubation period, and humans become infectious and able to transmit the virus to mosquitoes. The daily biting rate, extrinsic incubation period and probability of infection from human to mosquito or mosquito to human are temperature sensitive and can be illustrated as follow:

(1) The daily biting rate ($b$) of a female *Aedes aegypti* increased linearly with temperature for 21 °C < $T$ < 32 °C as described by *Scott, Amerasinghe & Morrison (2000)*:

$$b(T) = 0.0943 + 0.0043T.$$

(2) The probability of infection from human to mosquito per bite ($b_m$) can be described as:

$$b_m(T) = -0.9037 + 0.0729T,$$

for the temperature range 12.4 °C < $T$ < 26.1 °C and it is equal to 1 for 26.1 °C < $T$ < 32.5 °C (*Lambrechts, Paaijmansb & Fansiri, 2011*).

(3) The probability of transmission of the virus from an infected mosquito to human per bite ($b_h$) is a linear relation and it increases when 12.4 °C < $T$ < 28 °C decreases sharply when $T$ > 28 °C and equal to zero $T$ > 32.5 °C (*Lambrechts, Paaijmansb & Fansiri, 2011*). The following equation describes $b_h$:

$$b_h(T) = 0.001044T(T - 12.286)\sqrt{32.461 - T}.$$

(4) Extrinsic incubation period was demonstrated by *Focks, Daniels & Keesling (1995)* as a decreasing relationship to temperature by using an enzyme kinetics model for 12 °C < $T$ < 35 °C. We let $c$ denote the reciprocal of the extrinsic incubation period. It can be described by following equation:

$$c(T) = -0.1393 + 0.008T.$$

### *The mosquito's life cycle parameters*

The temperature also was an impact on the entomological parameters regarding the mosquito's life cycle. Depending on the temperature and the availability of food, *Aedes aegypti* can complete larval development in 4–7 days. The sensitivity of temperature can be described as follows:

(1) The mortality rate ($\mu_m$) of the mosquito *Aedes aegypti* was explored by *Yang, Macoris & Galvani (2009)* using the enzyme experiment. The results showed that the mortality

rate ranged form 0.027 to 0.092 per day as $10.54\,°C < T < 33.4\,°C$.

$$\mu_m(T) = 0.8692 - 0.159T + 0.01116T^2 - 3.408 \times 10^{-4}T^3 + 3.809 \times 10^{-6}T^4.$$

(2) The oviposition rate ($a$): *Yang, Macoris & Galvani (2009)* showed that the oviposition rate increases with temperature. The value was nearly zero where the temperature was $15\,°C$ and large values of $T > 30\,°C$.

$$a(T) = -15.837 + 1.2897T - 0.0163T^2.$$

(3) Pre-adult mosquito maturation rate ($s$) from egg to adult mosquitoes.

$$s(T,t) = (0.00483T - 0.00796)\left(p_s - c_s\cos\left(\frac{\pi t}{365} + \theta\right)\right)\Theta\left(p_s - c_s\cos\left(\frac{\pi t}{365} + \theta\right)\right).$$

$\theta$ is adjusted year cycle for $s(T,t)$. The Heaviside $\Theta$-function prevent $s(T,t)$ from becoming negative. This term is derived from *Coutinho et al. (2006)* and (*Yang, Macoris & Galvani, 2009*) to describe seasonal pre-adult mosquito maturation rate.

## The impact of rainfall on *Aedes aegypti* population

*Mogi et al. (1988)* examined the hatching rate and the amount of eggs in containers in Chiang Mai and reported that the peak of the population was approximately 1 month after the start of the rainy season, from June until the end of September. The egg population remained low in the dry season but increased exponentially during the first half of the rainy season, and then decreased sharply in the second half of the rainy season. Although the rain continued, the population of aquatic stage mosquito was actually decreasing as the food supply in containers declined and the competition among larval increased. The amount of rainfall is associated with the mosquito population by increasing breeding sites or egg carrying capacity. An equation of the population dynamics of *Aedes aegypti* is created. The egg-carrying capacity indicates the maximum population of aquatic mosquitoes (egg, larva, pupae) such that resources are sufficient and equation is as follows:

$$K(t) = \left(K_m + (1 - K_m)\sin^2\left(\frac{\pi t}{365} + \phi\right)\right)K_E \tag{1}$$

where $K(t)$ is the egg carrying capacity related to the amount of available food and space for eggs and then larvae will be able to develop, $K_m$ is fraction of the minimum egg carrying capacity in the area, $K_E$ is constant egg carrying capacity, $\phi$ is adjusted year cycle. $K_E$ is always positive because several containers are rainfall independent. To demonstrate the mosquito population under the influence of the rainfall and temperature, we set $K_E = 100,000$, $K_m = 0.18$ is chosen to be the ratio between the egg hatching rate in dry and rainy seasons in Chiang Mai (183:1,023) (*Mogi et al., 1988*).

## Vector–host dynamic models

The mosquito population is divided into 5 categories: infected aquatic stage, $I_E$, susceptible aquatic stage, $S_E$, susceptible adult mosquito, $S_M$, latent adult mosquito, $L_M$, and infectious

adult mosquito, $I_M$. Note that, the mosquito life span is too short to recover from dengue virus. The life of a mosquito consists of four stages: egg, larvae, pupae and adult stage. The first three stages live in the water and the last one stays in the air. In this study, we included the first three stages as aquatic stage and the last one as adult stage. The lifespan of each stage depends on several factors, for example, temperature, food supply and places for eggs hatching. The human population is classified into susceptible, $S_H$, infectious, $I_H$, and recovered individuals, $R_H$. Flows from the susceptible to infected classes of both populations depend on the biting rate of the mosquitoes, the transmission probabilities, as well as the number of infectious and susceptible individuals in each of the species. For this model, all the parameters with their respective values are summarized in Table 1. The model is described as following;

## Human compartment

The total human population, $N_H$, is $S_H + I_H + R_H$. The equations for the human compartment are the following:

$$\frac{dS_H}{dt} = \lambda_h N_H - \frac{x_1 b b_h I_M S_H}{N_H} - \mu_h S_H, \tag{2a}$$

$$\frac{dI_H}{dt} = \frac{x_1 b b_h I_M S_H}{N_h} - (\mu_h + r)I_H, \tag{2b}$$

$$\frac{dR_H}{dt} = (1 - \mu_d)r I_H - \mu_h R_H, \tag{2c}$$

$$\frac{dC_H}{dt} = \frac{x_1 b b_h I_M S_H}{N_h} \tag{2d}$$

where $C_H$ is the cumulative number of infection. It describes the total number of human infections during a given period.

## Adult mosquito compartment

The total population of adult mosquitoes, $N_M = S_M + L_M + I_M$. The equations are as follows:

$$\frac{dS_M}{dt} = s S_E - \frac{x_2 b b_m S_M I_H}{N_H} - \mu_m S_M, \tag{3a}$$

$$\frac{dL_M}{dt} = \frac{x_2 b b_m S_M I_H}{N_H} - (\mu_m + c)L_M, \tag{3b}$$

$$\frac{dI_M}{dt} = c L_M + s I_E - \mu_m I_M \tag{3c}$$

where $x_1$ and $x_2$ are the transmission factors which control the simulation results fit to the actual data. $x_1$ and $x_2$ in this study are the factors such that unaccounted in laboratory experiments. For example, mosquitoes bite the human for blood but fail to transmit the virus, or the humans live in the places that mosquito unable to contact. Also, the mosquito countermeasures are applied in the community.

**Table 1** The description of symbols in this study.

| Parameters | Meaning | Values |
|---|---|---|
| $\lambda_h$ | Human birth rate | 0.000044 |
| $\mu_h$ | Mortality rate of the humans | 0.00004 |
| $r$ | Recovery rate of the humans | 0.143 |
| $\gamma$ | Infection rate in mosquito's egg | 0.5 |
| $\mu_e$ | Mortality rate of the aquatic stage mosquito | 0.143 |
| $\mu_d$ | Death due to dengue | 0.001 |
| $\mu_m$ | Mortality rate of the mosquitoes | – |
| $a$ | Oviposition rate | – |
| $s$ | Pre-adult mosquito maturation rate | – |
| $b$ | Daily biting rate | – |
| $b_m$ | Probability of infection from human to mosquito per bite | – |
| $b_h$ | Probability of transmission of dengue virus from infected mosquitoes to humans per bite | – |
| $c$ | Inverse of extrinsic incubation period | – |
| $K$ | Egg-carrying capacity | – |
| $K_m$ | Fraction of minimum egg carrying capacity | 0.18 |
| $K_E$ | Egg-carrying capacity | 100,000 |
| $\phi$ | Adjusted year cycle for $K$ | $\pi/3$ |
| $\theta$ | Adjusted year cycle for $s$ | $\pi/12$ |
| $p_s$ | Climatic factor modulating winters | 0.55 |
| $c_s$ | Climatic factor modulating winters | 0.45 |
| $x_1$ | Transmission mosquito-human probability | 0.36 |
| $x_2$ | Transmission human-mosquito probability | 0.2 |
| $t$ | Time | – |
| $T$ | Mean daily temperature | – |
| $S_H$ | Susceptible human | – |
| $I_H$ | Infectious human | – |
| $R_H$ | Recovery human | – |
| $S_M$ | Susceptible adult mosquito | – |
| $L_M$ | Latent adult mosquito | – |
| $I_M$ | Infectious adult mosquito | – |
| $S_E$ | Susceptible aquatic stage mosquito | – |
| $I_E$ | Infectious aquatic stage mosquito | – |
| $z$ | Ratio between mosquito and human population | – |

**Notes.**
The symbol – means that the parameter is not constant value.

## Aquatic stage mosquito compartment

DENV can transfer from infected mosquitoes to eggs. The process is called vertical transmission (*Adams & Boots, 2010*). In this part, we divide the aquatic stage into susceptible and infectious population groups. The total population of aquatic mosquitoes (egg, larva and pupae), $N_E$, is $S_E + I_E$. The birth rate of the mosquitoes is described in term of logistic equations. We used the same assumption as *Coutinho et al. (2006)* that latent mosquitoes are unable to transmit dengue virus to theirs eggs. The populations of the

aquatic stage are as follows:

$$\frac{dS_E}{dt} = a\left(1 - \frac{S_E + I_E}{K}\right)(S_M + L_M + (1 - \gamma)I_M) - (s + \mu_e)S_E, \tag{4a}$$

$$\frac{dI_E}{dt} = a\left(1 - \frac{S_E + I_E}{K}\right)\gamma I_M - (s + \mu_e)I_E. \tag{4b}$$

The parameter, $\gamma$, represents the proportion of infected eggs (0.5 (*Coutinho et al., 2006*)) laid by infected female mosquitoes.

### An approximated threshold condition and the seasonal reproduction number

The seasonal reproduction number, $R_S$, is another form of the basic reproduction number as the climate factors are included. The value will alternate during the year and it can be calculated using the Van den Driessche and Watmough method (*Van den Driessche & Watmough, 2002*). The seasonal reproduction number is

$$R_S = \frac{\alpha}{2} + \sqrt{\frac{\alpha^2}{4} + \beta}, \tag{5}$$

where

$$\alpha = a\left(1 - \frac{N_E}{K_E}\right)\frac{sr}{(s + \mu_e)\mu_m},$$

$$\beta = \frac{x_1 x_2 b^2 b_h b_m zc}{\mu_m(\mu_m + c)(\mu_h + r)}$$

where $z$ is the ratio between the adult mosquito and human population. If $R_S$ is less than 1, the disease does not invade the population, and if $R_S$ is greater than 1, there is an outbreak of the disease (*Massad et al., 2011*). By not invading the population, the number of infected individuals always decreases in subsequent seasons of transmission. All calculations in this study were carried out by Matlab with ODE45 function for solving nonlinear equations. The objectives of this project are to evaluate the dengue incidences using the mathematical model and to estimate the effects of climate on the dengue transmission.

## RESULTS

### Number of dengue incidences; actual data vs. simulation results

The initial conditions $S_H(0)$, $I_H(0)$, and $R_H(0)$ were assumed to be 100,000, 25, and 0, respectively, in that the infectious population was estimated to be 5 cases in accordance with the number of confirmed cases in December. The initial values $S_M(0)$, $L_M(0)$, $I_M(0)$, $S_E(0)$, and $I_E(0)$ were set to be 10,000, 0, 0, 10,000, 100. We assumed the egg carrying capacity equal to human population. The parameters are as Table 1, temperature and season as illustrated above.

The data from our model is obtained from the average monthly dengue incidences during 2004–2014 in Chiang Mai. Figure 2 shows the dengue incidences generated by the model, the cumulative number of infection in that month, and the average monthly

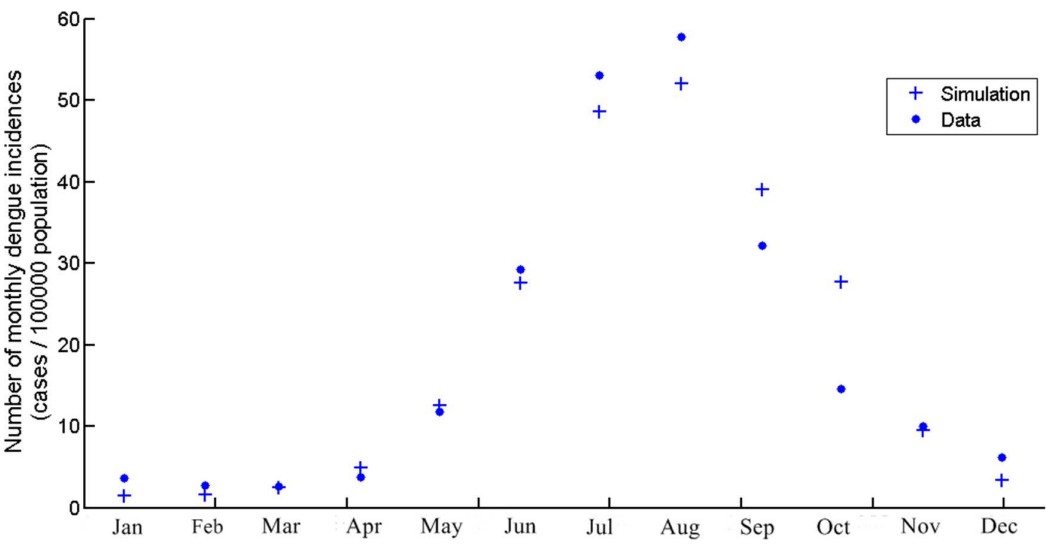

**Figure 2** **The number of monthly dengue incidences generated by the model and actual data of Chiang Mai from 2004 to 2014.** The scale is cases per 100,000 population.

incidences. The numbers are new infections in the month in question. The temperature is the temperature in Chiang Mai and seasonal factors as described in Table 1. The highest number of dengue incidences occurred in August. It is important to note that the asymptomatic dengue fever represents 75.7%–90.2% of total dengue infections (*Seyler, Grandesso & Le Strat, 2009*). In this study, we assumed that the actual infection number was 5 times greater than the reported cases, which is the same ratio as *Seyler, Grandesso & Le Strat (2009)*. However, the actual data showed only reported or symptomatic cases. Asymptomatic symptoms are generally ignored and unreported. In the model simulation, the infection is a combination of asymptomatic and symptomatic cases. Therefore, the number of infectious human generated by model was divided by 5 to represent only symptomatic cases and fit to the actual data.

## The mosquito population

Figure 3 illustrates the population of adult and infected mosquito by using model. The temperature used in this simulation was daily temperature of Chiang Mai and the seasonal factors also included. The running duration was one year ($t = 1$ is January, $1^{st}$). The simulations show the peak of mosquito population is 270,000 and infected mosquito population is 900. It implies that the highest ratio between mosquito and human population is 2.7. The time that both population groups successfully reach the highest number is $t = 157$ and $t = 247$ for mosquito and infected mosquito population respectively. After that the infected mosquito population start to declined and also the number of infected human population also reduce.

## Seasonal reproduction number

The optimal temperature for $R_S$ is calculated by using a constant ratio between the humans and the mosquitoes ($z$). Figure 4A illustrates $R_S$ as a function of the temperature (18 °C

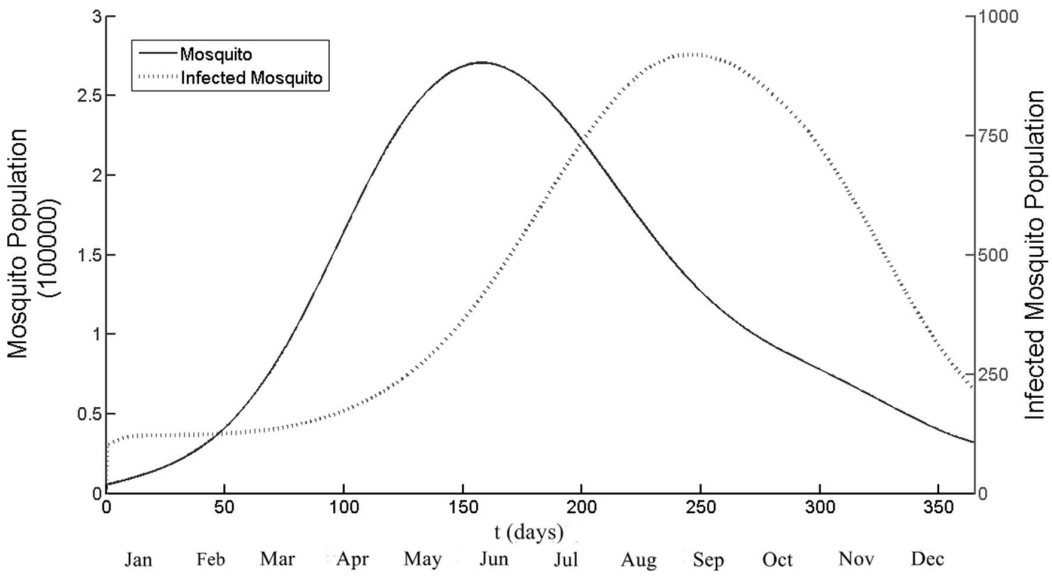

**Figure 3 The total number of mosquito and infected mosquito population generated by the model as a function of time (days).** $t = 1$ is January 1st and $t = 365$ is December 31st. The peak points are $t = 157$ and $t = 247$ for the mosquito and the infected mosquito population, respectively. Note that different scale, left side for mosquito and right side for infected mosquito population.

$< T < 32\,°\text{C})$ with constant value $z = 3$, which is above the highest number of results in Fig. 3. The function $R_S$ is increasing with the temperature when the temperature is from 20 to 29 °C. $R_S$ is over 1 when the temperature is between 24.9 and 31.7 °C. The highest values of $R_S$, 1.45, occurred at 28.9 °C, then decreasing rapidly thereafter. Figure 4B simulates the values $R_S$ as a function of time for a period of one year. The temperatures are the daily mean temperature in Chiang Mai. The value of $R_S$ reached its peak 1.43 in June ($t = 157$) and $R_S$ over 1 from $t = 82$ to $t = 238$.

## The impact of temperature to dengue incidences

The average yearly temperature in Chiang Mai is 26.7 °C. To investigate the impacts of changing mean temperature to the dengue incidences, the term $T_x$ was introduced. The new mean temperature is $T^* = T + T_x$. $T_x$ is increasing or decreasing temperature from mean value from 2004 to 2014. Figure 5 shows the total yearly dengue incidences as a function of $T_x$ and the ratio between the new total number of incidences and original number of incidences. The number of dengue incidences increases as mean temperature increases until $T_x > 2.1\,°\text{C}$ then the number decreases. The figure suggests that the highest incidence yielded at the mean temperature increased 1.1 °C equal to 293 cases per 100,000 population or an increase of 28.6% from average yearly temperature.

## DISCUSSION

Mathematical models are very useful tools in describing infectious diseases and in developing control strategies. We believe that the dynamic results were caused by the comprehensive effects of temperature-dependent parameters. In this study, we incorporated seven temperature-dependent and two seasonal parameters for the dengue

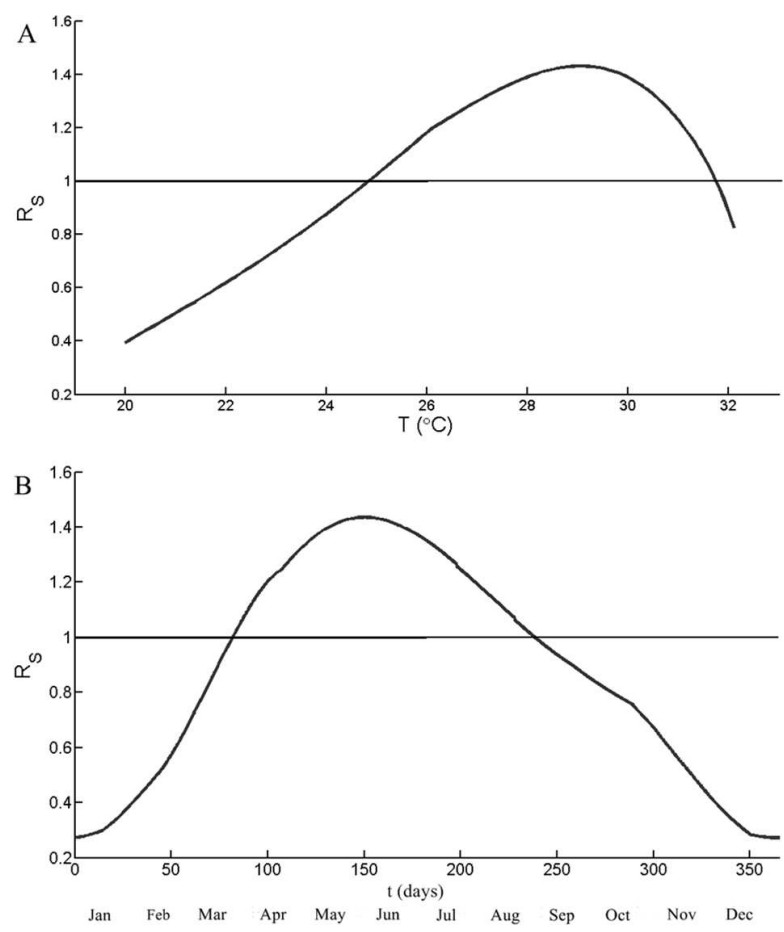

**Figure 4 Simulation results of the seasonal reproduction number ($R_S$).** (A) $R_S$ as a function of temperature and the ratio between mosquito and human, $z$, is 3. (B) $R_S$ as a function of time of period one year. The temperature is from the mean temperature of Chiang Mai during 2004–2014. The horizontal line indicates $R_S = 1$.

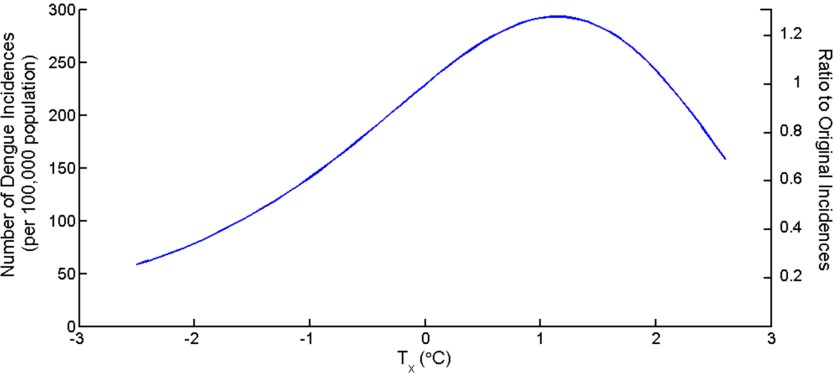

**Figure 5** The number of total incidences in one year as the temperature changes ($T_x$) from the mean temperature of Chiang Mai during 2004–2014 ($T_x = 0$).

transmission model and considered the potential outbreak of the dengue infection. The study showed the severity and persistence of the disease with seasonal fluctuation. The mosquito population dynamics were difficult to evaluate and most of the previous studies assumed that population remained unchanged. The climate factors, such as rainfall and temperature, will shorten or extend the life cycle of the mosquito population. The ratio between the mosquito and the human population ranges from 0.3 to 20 (*Chen & Hsien, 2012*). In this study, the mosquito population dynamics were set to be a function that corresponds to rainy season and life cycle of mosquito largely depend on temperature. After the start of the rainy season, the egg populations increased exponentially toward the peak abundance in the middle of the rainy season. The peak density tended to be higher in the rural area than in the urban area. This peak was followed by an exponential decline despite continuing rains and high temperatures. The seasonal pattern foreshadowed by one month the epidemic pattern of dengue infection (*Mogi et al., 1988*). It is believed that the population burst tended to consume all the food supply accumulated in the containers during the dry season, so the later generations were subjected to a more severe competition through food exploitation.

The reproduction number in this study is low compared to the other studies (*Nishiura, 2006*). The reason is that the transmission mosquito-human-mosquito factors ($x_1$, $x_2$) are introduced in order to fit the model to the actual data. The previous studies using transmission probability were based on the data from experiments. In reality, there are several factors that could increase or decrease transmission rate. In this study, we attempted to simulate the results to match the actual data of the dengue incidences.

The delay between the peak in the mosquito population and the peak in the dengue epidemic is nearly 90 days in this study as illustrated in Figs. 2 and 3. The mosquito population reach the peak in the early rainy season or around early June but the highest infection for mosquito and human occur months later in late August. This delay is due to the cyclic pattern of the mosquito population density. During unfavoured season, the mosquito population drops to a very low level with $R_S < 1$ (for transmission). In the beginning of the summer, the mosquito population begins to increase until it reaches a critical level at which $R_S$ is greater than 1 and then the transmission begins and required time to spread among both population groups, thus the delay occurs. This delay also mentioned in *Coutinho et al. (2006)* study.

Figure 4A have shown that the optimal temperature of dengue transmission is 28.9 °C, which is close to the mean temperature of Chiang Mai during the summer and the rainy season (27–29 °C). As shown in Fig. 4B, $R_S$ is larger than one when $t$ from 82 to 237. The disease starts to invade the population groups, the numbers of infected mosquitoes and humans are rising until reaching the peak and declining thereafter. The peak of $R_S$ occurred in the early June ($t = 157$) but the highest number of incidences occurred in August. The result is the same trend as the delay between mosquito population and infected population.

The impact of changing mean temperature also investigated as shown in Fig. 5. An increasing of 1 °C in the mean temperature causes an increasing of 28.1% in the number of the dengue incidences. Figure 5 illustrated that the number of dengue incidences in the hot

years may be as high as 4 times to cold years. *Wu, Lay & Guo (2009)* shows that with every 1 °C increase in the monthly average temperature in Taiwan, the total population at risk for dengue fever by 95%. The reason for this difference is that the mean temperature in Chiang Mai is close to the optimal temperature (28.9 °C) for several months. However, a very high temperature cause a high mortality rate and expand the extrinsic incubation period which results to mosquito to die before transmitting the virus to the humans.

Mathematical models have limit. For this study, The data shown two peaks in the rainfall number. The sinusoidal function may not the best solution to represent the mosquito population. However, the peak of aquatic mosquito population has only one peak (*Mogi et al., 1988*). The value of egg carrying capacity (*K*) is still unidentified. The assumption that *K* is equal to human population is widely used (*Chen & Hsien, 2012*). However, the number may be alternated for each year. The model of this study fit well with the number of average dengue incidences. Therefore, it can be used to evaluate the effects of climate to dengue transmission. Dengue transmission is also influenced by several factors that include rapid urbanization, capacity of public health systems, effectiveness of vector control systems, and the social behaviour of the population.

## CONCLUSION

The number of mosquito population varies according to seasonal variations. During favourable periods, when the size of the mosquito population increases, the potential for dengue infection in the humans also increases. The potential for the dengue transmission requires the following four factors: (1) a number of susceptible humans, (2) a number of mosquitoes, (3) virus transmission potential, and (4) a suitable climate.

The temperature plays a significant role in the dengue transmission and influences the dynamic modelling of vector–host interaction. The most suitable temperature for the dengue transmission is 28.7 °C. Rainfall provides breeding sites for the mosquitoes to hatch and develop into the adult stage. Both factors have a significant impact to the mosquito population and the dengue transmission dynamics. Our simulations confirm the impact of climate to dengue transmission is significant and suggest that the greatest potential of dengue transmission in Thailand occur in June to August which average temperature is 28–29 °C in the mid of rainy season. The mosquito population reach their peak three months before the peak of infection of mosquito and human. This study shows the impacts of climate especially temperature and seasonal variation to the transmission of dengue virus.

## ACKNOWLEDGEMENTS

The author would like to thank Prof. Raimo Nakki of the University of Jyväskylä, Finland, for carefully reading and commenting on the manuscript, as well as the anonymous reviewers for suggestions and comments.

### Funding

This study was funded by the Faculty of Science, Silpakorn University grant number SRF-PRG-2557-05. The funders had no role in study design, data collection and analysis, decision to publish, or preparation of the manuscript.

### Grant Disclosures

The following grant information was disclosed by the author:
Faculty of Science, Silpakorn University: SRF-PRG-2557-05.

### Competing Interests

The author declares there is no competing interests.

### Author Contributions

- Sittisede Polwiang conceived and designed the experiments, performed the experiments, analyzed the data, contributed reagents/materials/analysis tools, wrote the paper, prepared figures and/or tables, reviewed drafts of the paper.

### Supplemental Information

Supplemental information for this article can be found online at http://dx.doi.org/10.7717/peerj.1069#supplemental-information.

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
