# Peer review of "Retraction Notice"

_PeerJ, doi:10.7717/peerj.1069_

## Round 0.1 · original submission · Major Revisions

Please find here comments of the reviewers. Please provide your replies along with changes suggested by the reviewers, so as to improve the manuscript.

·

Basic reporting

• The introduction is fairly well done but I would appreciate if the author addressed a couple of minor issues:
o The last sentence of the first paragraph states that females obtain blood to gain energy for egg laying. While this is true, it would probably be more appropriate to replace “energy” with “nutrients” or say “energy and nutrients”.
o Towards the end of the second paragraph the author discusses the effects of “global warming” on dengue vectors. I would suggest replacing “global warming” with “climate change” because it is a more general term and encompasses possible precipitation changes which will undoubtedly influence vector population dynamics.

• This article does need extensive grammatical editing. I understand that English is not the author’s first language but often the grammatical errors make it difficult to read and understand. I suggest the author work with someone in the journal’s editing office or their university who speaks English as their first language.

Experimental design

Overall I think the author used an appropriate and interesting methodology. The equations used in the study seem appropriate and are justified by the literature. However, I do have a few questions and suggestions listed below:

• At the end of page 3 it reads, “…hatched pupae..” which is incorrect since larvae emerge from the eggs and complete 4 larval instar stages before becoming pupae and then finally adult mosquitoes.

• It should be made clear that c is actually the inverse of the EIP. When I first looked at equation 3b I thought it was incorrect until I realized the mistake was in how c was described. For instance, at 30°C c = 0.1. What this really means is that the EIP is 10 days at 30°C and the rate of completion of the EIP at 30°C is 0.1.

• Rainfall has a large impact on vector population dynamics and it is addressed in the paper using carrying capacity K, however, I would appreciate a little more detail describing K and its calculation. For instance, how are KE, KM, and the minimum egg carrying capacity determined? Additionally, there is very little information about φ and how it is calculated and what types of values it takes.

• Table 1 is very useful for looking up parameter names and values while reading through the methods section. I think this table could be expanded for other variables and parameters as well. I often found myself searching through the paper to remind myself of variable names, values, and equations (a, bm, bh, s, c, µm, Km, KE, NH, SH, IH, RH, NM, LM, I, M, NE, EE, IE). A separate table or expansion of Table 1 with these variable/parameters, their meanings, and their values/equations would be very helpful.

• In most instances the model was only run over a single year using averages. It might be worth also running the model under warmer/cooler and wetter/drier conditions. This could be done by using the lower/upper quartile temperature and/or precipitation for each month. This may yield interesting results.

• The model produces interesting results, however, no validation analysis has been performed or referenced. Is it possible to do such an analysis? If so, this would greatly enhance confidence in the results. It seems monthly temperature values were used as input but the model seems to be run using a daily time step. Is it possible to run the model using historical temperature data? If so, could the results be compared with dengue case data or mosquito population data from Chiang Mai?

• How were the initial values for the model results in Figure 3D chosen?

• In the second paragraph under the subtitle “The Effects of Climate to Dengue Transmission” is EM supposed to be LM?

Validity of the findings

• Under the section “Sensitive Analysis of Parameters” is says that vector control represents the only method to control outbreaks. I would say “vector control and vector-human contact reduction” or something similar because these are two different things but they are both discussed in the paper. The former refers to minimizing the vector population while the latter refers to avoiding contact with the vector.

• The author concludes through the sensitivity analysis that the best way to limit dengue transmission is through avoidance of the mosquitoes. This is because the model shows that the largest drop in dengue infections occurs when b is at half its value compared to when K is half its value or µm is doubled. However, this does not consider the difficulty in achieving the values. For instance, it might take a lot more time and money to reduce b by half than to reduce K by half. Assessing the effort to achieve these values is far beyond the scope of this study but it is worth mentioning some of the possible benefits and challenges of each of the intervention methods being modeled.

• The decline in the mosquito population later in the year is due to exhaustion of food supplies for the immature mosquitoes in the model. It seems to me, however, that the food supply would be constantly replaced by organic materials from leaves and other insects that fall into the habitat. Is there much evidence of food supplies being depleted later in the year? If so, what mechanism is replacing the food for the next year that is not replacing food throughout the year?

• The limitations section is rather brief and I believe there are some other limitation that should be mentioned. I understand, of course, that most can’t be avoided but it is worth discussing the limitations to put the results in perspective and prepare for future research. Some examples are listed below:
o No validation analysis was performed so it is difficult to assess the accuracy of the model results
o Much variability in the parameter values/equations exist but many are not well known or studied. I believe the author did their best to choose appropriate values but it should be acknowledge that there might be great variability in the values over time and space.
o The model was run using average monthly climate conditions but weather and climate can vary from day to day and year to year. This will not be reflected in the model results in the present study.

Additional comments

Overall I like the methodology of the paper and believe it has the potential to produce interesting and useful results. I do feel, however, that extensive grammatical editing needs to be done before I can fully judge the quality of the manuscript.

·

Basic reporting

This paper “The seasonal reproduction number of dengue fever: Impacts of climate to transmission” has a clear introduction and background and the relevant literature referenced.
Figures are relevant to the content of the article and structure follows the acceptable format.
The English needs substantial work to make it clear and readable both grammar and missing information. See colored marks and comments on the pdf file.

Experimental design

This study carried out original primary research within the Aims & Scope of the Journal. The research question on climate dependent dengue reproduction number is relevant and meaningful to the health field. The modelling is conducted rigorously under certain assumptions about reality. As usual, model is not reality but representation of certain aspects of reality. The methods are described with sufficient information after corrections to be made as pointed out in the attached pdf file. The research is modelling, more theory than experiment and thus is conducted in conformity with ethical standards.

Validity of the findings

The Author showed seasonal dependent Aedes aegypti population as a function of time based on both constant and monthly average of Chiang Mai, Thailand’s temperature over the recent 10 years. The result captured the one month delay of the adult mosquito’s population (peaked at 209 days) relative to the immature mosquito (peaked at 180 days). This seems similar to the dengue case profile. It is hard to compare when using “days” for the model’s output and “month” for the dengue cases. It would be easier to compare if the author converts all the time unit from days to months or show both units. This may be done by plotting Figure 2 in two time scales, such as, Figure 2A as it is now and Figure 2B using months for time.
The author showed the reproduction number Rs as a function of temperature and time – the seasonality and stated “The greatest potential of dengue transmission occurs at temperature equal to 28.7ºC. The seasonal reproduction numbers was 0.62-3.05, above unity from February to November and reached the peak in July.” This is very interesting. However, Figure 3B showed that the peak is in June, about 180 days. It is hard to understand why the peak of Rs does not match that of the adult vector population but the immature vector. Please recheck the calculation and discuss the reason for the differences.

Additional comments

Overall, this paper contains new results that are relevant in the dengue research field. Temperature dependent parameters have not been so widely used in the past in dengue modelling especially for the tropical dengue endemic areas. Therefore, giving the climate change situation, it is important to carry out this type of Mathematical modelling on dengue epidemic potential even for dengue endemic areas in order to support the effort on dengue early warning and policy for dengue vector control. The result is worth of publication after carefully checking for missing details, consistency, and English, especially answering the above point regarding Figure 3B.

---

## Round 0.2 · Minor Revisions

I recommend addressing current comments of both reviewers. It will improve quality of this MS.

·

Basic reporting

The quality of the grammar in the paper has improved, making it easier to read. There are, however, still many minor grammatical errors that should be fixed. I suggest an editor from the journal provide some assistance. The authors have sufficiently addressed the comments I made in the previous review.

Experimental design

The authors have clarified and provided more detail on the methods they employed in the study. Of particular note, they added an analysis of the sensitivity of annual dengue incidence to changing temperature. This is one of the most interesting parts of the paper. I would appreciate a little context for the range of values (-2.5 degrees c to +2.5 degrees c) chosen. For example, how does this compare to the actual variability in temperature and to projected climate change for the region.

Validity of the findings

In regards to the above comment, I believe a little more discussion about the results in Figure 5 is warranted. For example, is there a trend of warming temperatures in the region? If so, how much? What will the potential effects be using the information from Figure 5? For example, if temperatures are projected to increase from 0-2 degrees C, the model predicts that dengue fever cases will increase, however, they will decrease if temperatures increase much beyond 2 degrees C. Lastly, I believe the limitations of the model still need to be discussed (see previous review comments). They seem to be absent in this new version.

Additional comments

Overall I believe the manuscript has been greatly improved. It still needs some grammatical editing but reads better than before. The temperature change analysis was a great addition and should be discussed in a little more detail. Additionally, some of the model limitations should be acknowledged. This model seems promising and I hope the authors will continue to employ it in unique ways in future studies.

·

Basic reporting

The author has improved the article substantially both in language, clarity and content. New figures have added and some old figures have dropped. This has improved the quality of the research greatly. Figure 2 has given validity to your model when you can compare it with real data. Figure 3 and 5 give very interesting new information. Figure 4 is much more clear and useful than the original Figure for Rs.
Still little language improvement needed – see high lighted areas and some comments.
I would like to see some discussion on comparing the peak time between dengue incidences (Fig. 2) and the mosquito population (Fig.3). Different time unit makes it hard to compare. You may add a “day” scale in addition to “month” in Figure 2 or split Fig. 3 to (a – the current Fig. 3) and (b – Fig. 2 using day as the time unit) so that readers can compare the dynamics between mosquito population and the dengue incidence.

Experimental design

Good.

Validity of the findings

Yes.

Additional comments

Great job on the new version! Little work left to make it a good publication.

---

## Round 0.3 · accepted · Accept

Congratulations, the authors have carefully considered all major points of the reviewers and made appropriate changes, and so I declare that this manuscript is accepted for publication. Please make one final look for grammatical mistakes, so that any mistake found will be taken into consideration in the final publication.